# On-Treatment Albumin-Bilirubin Grade: Predictor of Response and Outcome of Sorafenib-Regorafenib Sequential Therapy in Patients with Unresectable Hepatocellular Carcinoma

**DOI:** 10.3390/cancers13153758

**Published:** 2021-07-26

**Authors:** Hung-Wei Wang, Po-Heng Chuang, Wen-Pang Su, Jung-Ta Kao, Wei-Fan Hsu, Chun-Che Lin, Guan-Tarn Huang, Jaw-Town Lin, Hsueh-Chou Lai, Cheng-Yuan Peng

**Affiliations:** 1Center for Digestive Medicine, Department of Internal Medicine, China Medical University Hospital, Taichung 404, Taiwan; D29280@mail.cmuh.org.tw (H.-W.W.); D12936@mail.cmuh.org.tw (P.-H.C.); D13845@mail.cmuh.org.tw (W.-P.S.); D16666@mail.cmuh.org.tw (J.-T.K.); D28991@mail.cmuh.org.tw (W.-F.H.); D83949@mail.cmuh.org.tw (C.-C.L.); D96790@mail.cmuh.org.tw (G.-T.H.); D8649@mail.cmuh.org.tw (J.-T.L.); 2School of Medicine, China Medical University, Taichung 404, Taiwan; 3School of Chinese Medicine, China Medical University, Taichung 404, Taiwan

**Keywords:** albumin-bilirubin (ALBI) grade, hepatocellular carcinoma, sorafenib–regorafenib sequential therapy

## Abstract

**Simple Summary:**

Regorafenib after sorafenib therapy improved survival in patients with advanced hepatocellular carcinoma in the RESORCE study. The aim of our retrospective study was to investigate the predictors of response and outcome of regorafenib therapy in patients with unresectable hepatocellular carcinoma in whom sorafenib therapy had failed. We demonstrated that albumin-bilirubin grade at the initiation of regorafenib therapy is an independent predictor of disease control, progression-free survival, and overall survival. Patients with albumin-bilirubin grade 2 and an alpha-fetoprotein level of ≥20 ng/mL had the worst progression-free survival (after regorafenib therapy) and overall survival (after regorafenib and sorafenib therapy). Thus, a combination of albumin-bilirubin grade and alpha-fetoprotein level can be used to stratify patients with unresectable hepatocellular carcinoma by progression-free survival and overall survival probability for sorafenib–regorafenib sequential therapy.

**Abstract:**

In the RESORCE study, regorafenib after sorafenib therapy improved survival in patients with advanced hepatocellular carcinoma (HCC). In total, 88 patients with unresectable HCC who received sorafenib–regorafenib sequential therapy were enrolled. The objective response rate and disease control rate were 19.3% and 48.9%, respectively, for regorafenib therapy (median duration: 8.1 months). Median progression-free survival (PFS) after regorafenib therapy was 4.2 months (95% CI: 3.2–5.1). The median overall survival (OS; from initiation of either sorafenib or regorafenib) was not reached in this cohort. According to multivariate Cox regression analyses, albumin-bilirubin (ALBI) grade at the initiation of regorafenib therapy is an independent predictor of disease control, PFS, and OS. Moreover, the combination of ALBI grade 2 and an alpha-fetoprotein (AFP) level of ≥20 ng/mL was an independent predictor of PFS (hazard ratio (HR): 3.088, 95% CI: 1.704–5.595; *p* < 0.001) for regorafenib therapy, and OS for both regorafenib (HR: 3.783, 95% CI: 1.316–10.88; *p* = 0.014) and sorafenib–regorafenib sequential (HR: 4.603, 95% CI: 1.386–15.29; *p* = 0.013) therapy. A combination of ALBI grade and AFP level can be used to stratify patients with unresectable HCC by PFS and OS probability for sorafenib–regorafenib sequential therapy.

## 1. Introduction

Multiple lines of therapy can improve overall survival (OS) in cancer. For advanced hepatocellular carcinoma (HCC), the median OS increases from 11–14 months to 35 months after two or more lines of systemic therapy [1,2,3]. Several treatment options are available after first-line therapy for advanced HCC fails, including tyrosine kinase inhibitors (TKIs) and immune checkpoint inhibitors [4,5,6,7]. However, information regarding the survival benefit of sequential therapy and its optimal regimen is limited [8,9]. In the RESORCE study, regorafenib was first demonstrated to have survival benefits, exhibiting a hazard ratio (HR) of 0.63 versus placebo after failure of sorafenib therapy; regorafenib was subsequently approved for second-line treatment of patients with advanced HCC [1,10,11]. Nonetheless, it remains unclear who may benefit most from sorafenib–regorafenib sequential therapy. Identification of potential responders may guide the selection of optimal patients for sorafenib–regorafenib sequential therapy in advance.

Liver reserve may worsen during anticancer therapy, and preserving liver function is essential to achieving favorable outcomes of sequential systemic therapy [3]. The albumin–bilirubin (ALBI) grade, as well as the Child–Pugh score, reflects liver reserve and can serve as a predictive factor in patients with HCC undergoing anticancer treatment [12,13,14]. Takada et al. demonstrated that highly preserved liver function (Child–Pugh A5 or ALBI grade 1) after sorafenib treatment failure is beneficial for second and third-line sequential therapy in patients with unresectable HCC [15]. For sorafenib-treated HCC patients, baseline modified ALBI grade in combination with Child–Pugh score might be a predictor of response to second-line sequential therapy, including regorafenib or ramucirumab [16]. Alpha-fetoprotein (AFP) is recognized as a serum marker of HCC and is related to tumor burden in some patients. In a REACH-II study, an AFP level of ≥400 ng/mL could be used as a selection criterion for sequential therapy with ramucirumab [17]. Moreover, on-treatment AFP changes may predict the treatment response. Given the current knowledge gap in the systemic therapy of HCC, we tried to investigate predictors of regorafenib therapy response and survival outcome in patients with unresectable HCC in whom sorafenib therapy had failed.

## 2. Materials and Methods

### 2.1. Patient Recruitment and Definitions

Our study was a retrospective cohort study of patients from a single tertiary care medical center in Taiwan. In total, 88 patients with unresectable HCC who had received sorafenib–regorafenib sequential therapy, were enrolled from September 2012 to July 2020. Furthermore, we enrolled another cohort (*n* = 90) of patients with unresectable HCC who did not receive regorafenib therapy after sorafenib treatment failure from September 2012 to September 2019 as an historical control group for comparing survival. Baseline and on-treatment clinical characteristics (including age, sex, Eastern Collaborative Oncology Group performance status, Child–Pugh score, and Barcelona Clinic Liver Cancer (BCLC) stage), laboratory data (including albumin, aspartate aminotransferase (AST), alanine aminotransferase (ALT), total bilirubin, and AFP levels; international normalized ratio, and platelet count), and imaging findings were collected. Contrast-enhanced dynamic computed tomography or magnetic resonance imaging was scheduled for patients every 2–3 months to assess treatment response. Tumor response to regorafenib therapy was evaluated with the modified Response Evaluation Criteria in Solid Tumors (mRECIST) [18]. The Fibrosis-4 (FIB-4) and ALBI scores were calculated according to the following formulas: FIB-4 = (age [years] × AST [U/L])/(platelet count [10^9^/L] × ALT level [U/L]^1/2^) [19], and ALBI score = [Log_10_ bilirubin level (μmol/L) × 0.66] + [albumin level (g/L) × −0.085] [12]. An FIB-4 cutoff value of > 3.25 predicted advanced fibrosis with a specificity of 97% and a positive predictive value of 65% [19]. ALBI grades 1, 2, and 3 respectively correspond to ALBI scores of ≤−2.60, −2.60 to −1.39, and >−1.39 [12]. This study was conducted in accordance with the 1975 Declaration of Helsinki. Patient informed consent was waived because each identification number was encrypted to protect their privacy, and the study was approved by the Research Ethics Committee of China Medical University Hospital (CMUH110-REC3-027).

### 2.2. Statistical Analysis

The categorical data of the two groups were compared using Fisher’s exact test, as applicable. Continuous data were evaluated for normality using the Kolmogorov–Smirnov test. Continuous data, which are expressed as medians and interquartile ranges, were analyzed using the Mann–Whitney U test. We used the area under the receiver operating characteristic curve (AUROC) to evaluate the performance of AFP values in predicting progression-free survival (PFS) and OS. The optimal cutoff value of AFP (17.3 ng/mL) was determined according to the Youden index. For convenience, we adopted the popular integer value (20 ng/mL) as the cutoff value of AFP for further analysis. OS and PFS were determined using Kaplan–Meier curves, and univariate analysis with the log-rank test was used to compare groups. Child–Pugh score and ALBI grade were used as covariates and were considered as confounding factors. Therefore, we adopted three models (based on Child–Pugh score, ALBI grade, and a combination of ALBI grade and AFP level) for multivariate Cox regression or logistic regression analysis. The HRs for survival predictors were determined using univariate and multivariate Cox regression analyses. Odds ratios for predictors of treatment response were determined through univariate and multivariate logistic regression analyses. A stepwise multivariate analysis was performed with variables whose *p* values were <0.25 in the univariate analyses. Statistical analyses were performed using SPSS version 26.0 (IBM Corp., Armonk, NY, USA). A *p* value of <0.05 was considered statistically significant.

## 3. Results

### 3.1. Baseline and On-Treatment Characteristics

The median age of the study group before regorafenib therapy was 66 ± 14 years. Among all patients, 69 (78.4%) were men, 83 (94.3%) were classified as Child–Pugh class A (score: 5 or 6), and 64 (72.7%) had BCLC stage C disease. In total, 19 patients (21.6%) died during the follow-up period, which had median durations of 18 and 8.1 months for sorafenib–regorafenib sequential therapy and regorafenib therapy, respectively. The variables at baseline (before sorafenib therapy) and during treatment (before regorafenib therapy) are presented in Table 1. The median treatment durations were 5.2 and 2.8 months for sorafenib and regorafenib, respectively. The median interval from sorafenib failure to regorafenib therapy was 1.4 months. Appendix A shows the baseline and on-treatment liver reserve during regorafenib therapy. Among patients with Child–Pugh class A, 54.5–68% maintained their liver reserve and could receive the next round of regorafenib therapy, as per Taiwan’s National Health Insurance guidelines. Because of disease progression, the majority of patients (85.7–100%) did not have the next round of regorafenib therapy approved.

To assess the treatment efficacy of sequential therapy with regorafenib, we enrolled an historical cohort of patients who had received sorafenib therapy alone, without sequential systemic treatment, after progressive disease (PD). The characteristics of these two cohorts were compared before and after propensity-score matching (Appendix A). Approximately 56.7% and 55.7% of the cohorts receiving and not receiving sequential therapy respectively received locoregional therapy, including radiofrequency ablation, transarterial chemoembolization, and radiotherapy, for palliative tumor control after sorafenib failure. The 6-month OS rates for patients receiving and not receiving regorafenib sequential therapy were 89% and 35.9%, respectively. Patients receiving regorafenib sequential therapy had significantly higher median OS compared with those without regorafenib therapy (Figure 1a, log-rank test, *p* < 0.001). After propensity-score matching, patients receiving regorafenib sequential therapy exhibited significantly higher OS than those not receiving regorafenib therapy (Figure 1b, log-rank test, *p* < 0.001). Furthermore, the median progression-free survival (PFS) for regorafenib therapy was 4.2 months (95% confidence interval (CI): 3.2–5.1), and a median OS was not reached (Figure 1c). A median OS for sorafenib–regorafenib sequential therapy was also not reached (Figure 1d).

### 3.2. Response to Sequential Therapy with Regorafenib

Treatment response was evaluated according to the mRECIST [18]. For all patients (*n* = 88), the objective response rate and disease control rate (DCR) were 19.3% and 48.9%, respectively. Table 2 presents the treatment response rates for patients stratified by Child–Pugh score or ALBI grade. The subgroup with a Child–Pugh score of 5 exhibited a higher DCR than the subgroup with a score of 6 (55.9% vs. 33.3%, *p* = 0.089). The subgroup with ALBI grade 1 exhibited a significantly higher DCR than did the subgroup with ALBI grade 2 (64.6% vs. 30%, *p* = 0.002).

### 3.3. On-Treatment Factors Associated with PFS after Regorafenib Therapy

We investigated factors before the initiation of regorafenib therapy that were predictive of PFS. According to the univariate Cox regression analyses, the Child–Pugh score (5 vs. 6), ALBI grade (1 vs. 2), combination of ALBI grade 2 and an AFP level of ≥20 ng/mL (yes vs. no), and AST level, were associated with PFS (Appendix A). According to multivariate Cox regression analyses, Child–Pugh score (5 vs. 6) and AFP level (<20 vs. ≥20 ng/mL) were not independent predictors in the model based on the Child–Pugh score, whereas sex (male vs. female), ALBI grade (1 vs. 2) and AFP level (<20 vs. ≥20 ng/mL) were independent predictors in the model based on ALBI grade. The combination of ALBI grade 2 and an AFP level of ≥20 ng/mL was an independent predictor of PFS in the model based on both ALBI grade and AFP level (Table 3, HR: 3.088, 95% CI: 1.704–5.595, *p* < 0.001). The combination of ALBI grade and AFP level could be used to stratify patients by the probability of 6-month PFS (Figure 2a, log-rank test, *p* = 0.001). The median PFS was 2.3 (95% CI: 1.8–2.8) and 5.6 months (95% CI: 3.0–8.3), respectively, for patients with both ALBI grade 2 and an AFP level of ≥20 ng/mL and for all the other subgroups combined.

### 3.4. On-Treatment Factors Associated with Overall Survival (OS) after Regorafenib Therapy

We investigated factors before the initiation of regorafenib therapy that were predictive of OS. The multivariate Cox regression analysis revealed that the combination of ALBI grade 2 and an AFP level of ≥20 ng/mL independently predicted OS in the model based on the combination of ALBI grade and AFP level (Appendix A, Table 4; HR: 3.783, 95% CI: 1.316–10.88, *p* = 0.014). The combination of ALBI grade and AFP level could be used to stratify patients by the probability of 6-month OS (Figure 2b, log-rank test, *p* = 0.001). The 6-month OS rates for patients with both ALBI grade 2 and an AFP level of ≥20 ng/mL and for the other subgroups combined were 78.6% and 92.5%, respectively. The median OS for patients with ALBI grade 2 and an AFP level of ≥20 ng/mL was 10.0 months (95% CI: 7.1–12.9).

### 3.5. On-Treatment Factors Associated with OS after Sorafenib-Regorafenib Sequential Therapy

We explored whether factors before the initiation of regorafenib therapy were predictive of OS after sorafenib–regorafenib sequential therapy. Multivariate Cox regression analyses revealed ALBI grade (1 vs. 2), AFP level (<20 vs. ≥20 ng/mL), and the combination of ALBI grade and AFP level to be independent predictors in two models (Table 5). Moreover, the subgroup with ALBI grade 2 and an AFP level of ≥20 ng/mL exhibited a significantly lower probability of 2-year OS than the other subgroups combined, with the median OS being 26.0 months (95% CI: 13.2–38.7; Figure 3, log-rank test, *p* < 0.001).

### 3.6. On-Treatment Factors Associated with Response to Regorafenib Therapy

Multivariate analysis identified sex and ALBI grade as predictors of disease control (including complete response, partial response, and stable disease) after regorafenib therapy (*p* = 0.039 and 0.006, respectively). However, no factor was significantly predictive of objective response (complete or partial response; Appendix A). The subgroup with PD had a median OS of 10 months (95% CI: 6.9–13.2), which was significantly lower than that of the subgroup achieving disease control (Figure 4, log-rank test, *p* < 0.001).

### 3.7. Effect of Sorafenib–Regorafenib Sequential Therapy on OS

In order to further evaluate the effect of the sequential therapy, we combined these two patient cohorts with or without sequential therapy (*n* = 178) to investigate the predictors of OS. In the ALBI grade-based model, ALBI grade (1 vs. 2), AFP (<20 vs. ≥20 ng/mL), sequential therapy (yes vs. no) and locoregional therapy (yes vs. no) were independent predictors for OS after failure of sorafenib therapy (Appendix A). In the combined ALBI and AFP-based model, ALBI grade 2 and AFP ≥ 20 ng/mL (yes vs. no), sequential therapy (yes vs. no) and locoregional therapy (yes vs. no) were independent predictors for OS after failure of sorafenib therapy (Appendix A). To eliminate the confounding effect of locoregional therapy, the subgroup of patients without receiving locoregional therapy after failure of sorafenib therapy (*n* = 78) was selected to analyze the effect of the sequential therapy. For OS after failure of sorafenib therapy, AFP (<20 vs. ≥20 ng/mL) and sequential therapy (yes vs. no) were independent predictors in the ALBI grade-based model. ALBI grade 2 and AFP ≥ 20 ng/mL (yes vs. no), sequential therapy (yes vs. no) were independent predictors in the combined ALBI and AFP-based model (Appendix A). For OS after sorafenib therapy, sequential therapy (yes vs. no) was also an independent predictor. Thus, sequential therapy was an independent predictor of OS not only for the entire cohort, but also for the subgroup without receiving locoregional therapy after failure of sorafenib therapy (Appendix A).

### 3.8. Association of Adverse Event Profile with Sorafenib-Regorafenib Sequential Therapy

Table 6 shows the incidence and severity of adverse events during sequential therapy. Hand-foot skin reaction (HFSR), diarrhea, and hypertension were the three most common events of any grade during both sorafenib therapy (59.1%, 36.4%, and 14.8%, respectively) and regorafenib therapy (31.8%, 38.6%, and 4.5%, respectively). The incidence and severity of HFSR decreased during the treatment course. We further explored the possible relationship between the HFSR experienced by patients during sorafenib–regorafenib sequential therapy (Table 6). Among the 36 patients who did not experience HFSR during sorafenib therapy, 7 (19.4%) experienced HFSR during regorafenib therapy. Nonetheless, for patients with any HFSR during sorafenib therapy, the incidence and severity of HFSR decreased during regorafenib therapy.

## 4. Discussion

In the RESORCE study, regorafenib therapy demonstrated a DCR of 65.7% and improved survival (median OS of 26 months with sorafenib–regorafenib therapy) in patients with advanced HCC that progressed after sorafenib therapy [11]. In an interim analysis of the observational REFINE trial (*n* = 498), patients who received regorafenib therapy had median PFS and OS of 3.7 and 13.2 months, respectively, which are consistent with the results of the RESORCE study [20]. Three recent real-world Asian studies involving 38, 44, and 305 patients reported a median PFS and OS of 3.1–6.9 and 12.1–17.3 months, respectively [21,22,23]. The present study demonstrated a longer median PFS (4.2 months) and OS (not reached) than did the RESORCE and REFINE studies. Moreover, we employed propensity-score matching for liver function (Child–Pugh score and ALBI grade) and tumor status (BCLC stage and AFP level) to compare the OS benefit among patients receiving or not receiving regorafenib therapy (Appendix A). We demonstrated that sequential therapy with regorafenib prolonged survival in patients who had failed sorafenib therapy (Figure 1b).

Although the Child–Pugh classification is the most common scoring system for liver function, the ALBI grade is more objective because its formula contains only two variables, which are easily measurable: serum albumin and total bilirubin. In the present study, we demonstrated that the on-treatment ALBI grade was a predictor of treatment response (Table 2 and Appendix A) and PFS after regorafenib therapy and of OS after sorafenib–regorafenib sequential therapy (Table 3, Table 4 and Table 5). Good liver reserve and low AFP levels are independently associated with longer median OS after sorafenib therapy [24]. Furthermore, the changes in AFP level during treatment may be predictive of an early response to sorafenib therapy [25,26,27]. No study has reported an association between the baseline AFP level or changes in AFP level and survival after regorafenib sequential therapy. In this study, the AFP level at the initiation of regorafenib therapy, but not on-treatment changes in AFP level, was associated with treatment outcome. Therefore, we combined the ALBI grade and AFP level at the initiation of regorafenib therapy. The combination of ALBI grade 2 and an AFP level of ≥20 ng/mL was an independent negative predictor of PFS (HR: 3.088, 95% CI: 1.704–5.595, *p* < 0.001) and OS (HR: 3.783, 95% CI: 1.316–10.88, *p* = 0.014) and can be implemented to predict treatment outcome in patients before regorafenib initiation.

The incidence of drug-related treatment-emergent adverse events (TEAEs), including HFSR, diarrhea, and hypertension, was similar for sorafenib and regorafenib. This finding is consistent with that of the phase 3 RESORCE trial [28]. In our study, most patients experienced less severe HFSR during regorafenib therapy than they did during sorafenib therapy (Table 6), possibly because patients had already developed tolerance to the skin toxicity of sorafenib and gained experience in performing skin care, which was useful for preventing HFSR when they were exposed to regorafenib. A meta-analysis revealed the dermatologic adverse events to be positively correlated with survival for first-line systemic therapy [29]. However, we did not identify HFSR as an independent factor of response to regorafenib therapy (Appendix A). Further investigation is warranted to clarify the role of HFSR in predicting the response to regorafenib therapy.

Our study is clinically relevant. First, it was a retrospective cohort study in a real-world setting, investigating the survival benefit of regorafenib therapy. Second, the combination of ALBI grade and AFP level at the initiation of regorafenib therapy can be used to stratify patients by their survival probability from receiving sequential therapy. Third, many patients received other therapy in real-world settings, such as locoregional therapy or radiotherapy, in addition to sequential therapy; such additional therapy might have partly accounted for the treatment efficacy. Nonetheless, similar proportions of patients from the entire cohort and from the propensity-score-matched cohort who did not receive regorafenib therapy after sorafenib failure received palliative locoregional therapy, suggesting the survival benefit of regorafenib therapy. Moreover, we demonstrated that sequential therapy was an independent predictor of OS not only for the entire cohort, but also for the subgroup, without receiving locoregional therapy after failure of sorafenib therapy (Appendix A). Fourth, we investigated TEAEs, in particular the incidence and severity of HFSR, during the course of sorafenib–regorafenib sequential therapy, thereby providing useful information for HFSR management during HCC therapy.

The study has some limitations. First, this was a retrospective study with median follow-up periods of 18 and 8.1 months for sorafenib–regorafenib sequential therapy and regorafenib therapy, respectively. A longer follow-up was needed to determine the median OS in our cohort. Second, we could not guarantee the daily dose of regorafenib during treatment because adjustments of the TKI dose by patients might not have been accurately documented in the medical records. Third, the two cohorts were enrolled respectively before and after reimbursement of regorafenib therapy began in Taiwan. Although we compared these two cohorts after propensity-score matching, selection bias due to the Will Rogers phenomenon might have occurred [30].

## 5. Conclusions

The ALBI grade at the initiation of regorafenib therapy is an independent predictor of disease control, PFS, and OS. A combination of ALBI grade and AFP level can be used to stratify patients with unresectable HCC by probability of PFS and OS for sorafenib–regorafenib sequential therapy. Our findings may provide a guide to clinicians in identifying the optimal candidates for regorafenib therapy after failure of sorafenib therapy in patients with unresectable HCC.

## Figures and Tables

**Figure 1 cancers-13-03758-f001:**
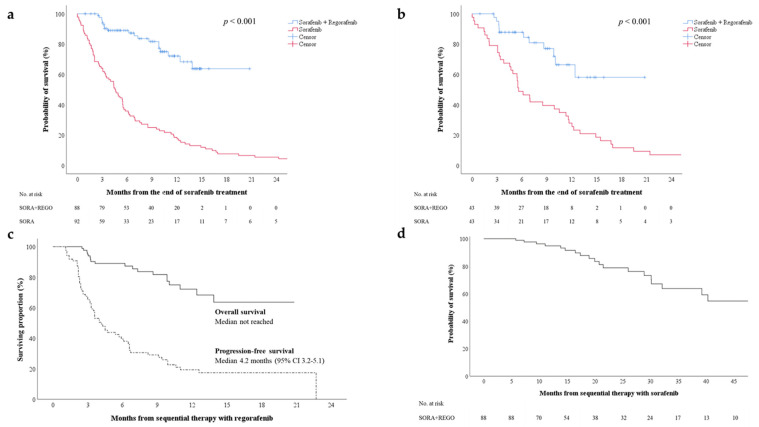
(**a**) Overall survival after initiation of regorafenib therapy for two cohorts receiving or not receiving regorafenib sequential therapy (sorafenib + regorafenib vs. sorafenib alone); (**b**) overall survival for the cohorts after propensity-score matching; (**c**) progression-free and overall survival after initiation of regorafenib therapy; (**d**) overall survival after initiation of sorafenib therapy. CI—confidence interval; REGO—regorafenib; SORA—sorafenib.

**Figure 2 cancers-13-03758-f002:**
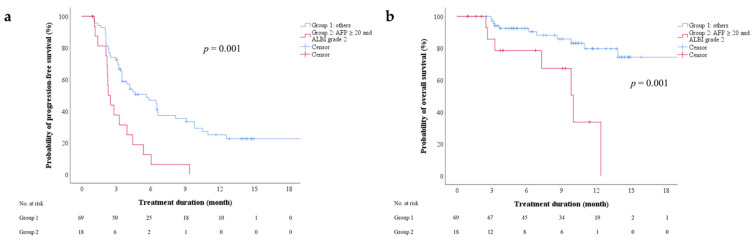
(**a**) Progression-free and (**b**) overall survival after initiation of regorafenib therapy, stratified by combination of AFP level and ALBI grade. AFP—alpha-fetoprotein; ALBI—albumin-bilirubin.

**Figure 3 cancers-13-03758-f003:**
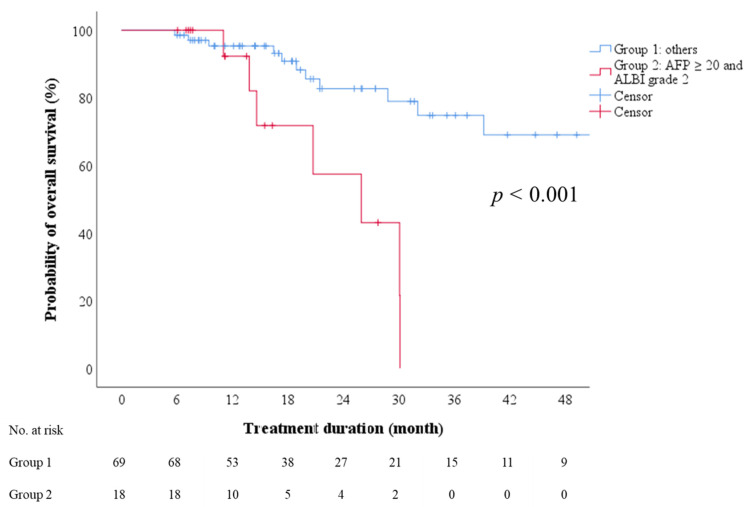
Overall survival after initiation of sorafenib therapy, stratified by combination of AFP and ALBI grade. AFP—alpha-fetoprotein; ALBI—albumin-bilirubin.

**Figure 4 cancers-13-03758-f004:**
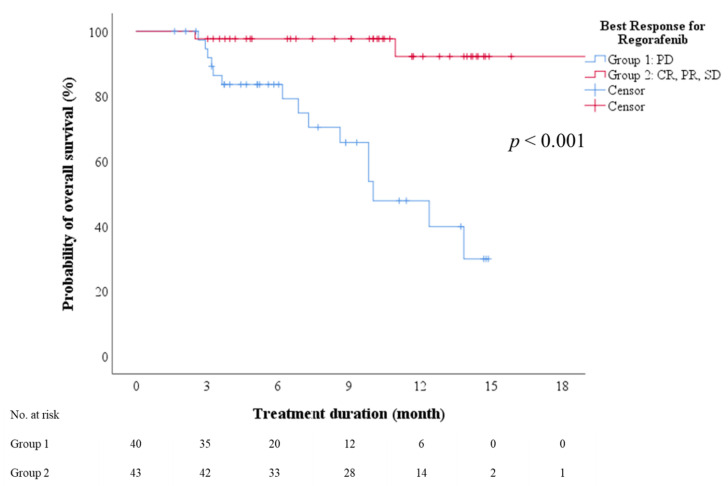
Overall survival after initiation of regorafenib therapy stratified by best treatment response. CR—complete response; PD—progressive disease; PR—partial response; SD—stable disease.

**Table 1 cancers-13-03758-t001:** Characteristics of patients receiving sorafenib–regorafenib sequential therapy.

Variables Median ± IQR or *n* (%)	All Patients (*n* = 88)	Baseline (Before SORA)	On-Treatment (Before REGO)
Age (year)		65 ± 15	66 ± 14
Sex	Male/female	69/19	
Etiology	HBV/HCV/ HBV+HCV/NBNC	45/28/2/13	
ECOG PS	0/1/2	84/4/0	75/10/3
Child–Pugh score	5/6/≥7	74/11/3	59/24/5
Child–Pugh class	A/B/C	85/3/0	83/5/0
ALBI score		−2.71 ± 0.60	−2.64 ± 0.57
ALBI grade	1/2/3	54/33/0	48/40/0
FIB-4		2.54 ± 2.07	3.59 ± 3.13
FIB-4	<3.25/≥3.25	45/29	31/45
BCLC stage	A/B/C	1/23/64	1/23/64
MVI	presence	24 (27.3)	28 (31.8)
EHS	presence	44 (50)	51 (58)
AFP (ng/mL)		10.7 ± 118.5	17.3 ± 292.4
AFP (ng/mL)	<400/≥400	72/13	66/21
Albumin (g/dL)		4.1 ± 0.7	4.0 ± 0.7
AST (U/L)		40 ± 34	39 ± 21
ALT (U/L)		36 ± 32	31 ± 22
Total bilirubin (mg/dL)		0.8 ± 0.4	0.8 ± 0.5
INR		1.04 ± 0.10	1.05 ± 0.10
TKI duration (months)		5.2 ± 9.8	2.8 ± 3.4

Abbreviations: AFP—alpha-fetoprotein; ALBI—albumin–bilirubin; ALT—alanine aminotransferase; AST—aspartate aminotransferase; BCLC—Barcelona Clinic Liver Cancer staging; ECOG PS—Eastern Cooperative Oncology Group performance status; EHS—extrahepatic spread; FIB-4—fibrosis index based on four factors; HBV—hepatitis B virus; HCV—hepatitis C virus; INR—international normalized ratio; IQR—interquartile range; MVI—macrovascular invasion; NBNC—non–hepatitis B and non–hepatitis C; PLT—platelets; REGO—regorafenib; SORA—sorafenib; TKI—tyrosine kinase inhibitor.

**Table 2 cancers-13-03758-t002:** Treatment response to sequential therapy with regorafenib (*n* = 88).

Evaluable Response	All Patients (*n* = 88)	Child–Pugh A5 (*n* = 59)	Child–Pugh A6 (*n* = 24)	ALBI Grade 1 (*n* = 48)	ALBI Grade 2 (*n* = 40)
Best Response, *n* (%)					
CR	3 (3.4)	2 (3.4)	1 (4.2)	3 (6.3)	0 (0)
PR	14 (15.9)	11 (18.6)	2 (8.3)	8 (16.7)	6 (15)
SD	26 (29.5)	20 (33.9)	5 (20.8)	20 (41.7)	6 (15)
PD	40 (45.5)	24 (40.7)	14 (58.3)	16 (33.3)	24 (60)
Non-assessable	5	2	2	1	4
ORR	17 (19.3)	13 (22.0)	3 (12.5)	11 (22.9)	6 (15)
DCR	43 (48.9)	33 (55.9)	8 (33.3)	31 (64.6)	12 (30)
For Responders					
Time to response (days)	75 (7–357)	94 (7–357)	49 (49–59)	117 (7–357)	64 (14–94)

Abbreviations: ALBI—albumin-bilirubin; CR—complete response; DCR—disease control rate; ORR—objective response rate; PD—progressive disease; PR—partial response; SD—stable disease.

**Table 3 cancers-13-03758-t003:** Results of multivariate Cox regression analyses of predictors of PFS after regorafenib therapy.

	ALBI Grade-Based Model		Combined ALBI and AFP-Based Model	
Variables	Multivariate Hazard Ratio (95% CI)	*p* Value	Multivariate Hazard Ratio (95% CI)	*p* Value
Age (year)				
Male vs. female	2.046 (1.050–3.984)	0.035	1.870 (0.965–3.624)	0.064
ALBI grade 1 vs. 2	0.432 (0.258–0.722)	0.001		
FIB-4 < 3.25 vs. ≥3.25				
BCLC stage B vs. C				
MVI (no vs. yes)				
EHS (no vs. yes)				
AFP (ng/mL) <20 vs. ≥20 (ng/mL)	0.556 (0.337–0.919)	0.022		
ALBI grade 2 and AFP ≥ 20 ng/mL (yes vs. no)			3.088 (1.704–5.595)	<0.001
Albumin (g/dL)				
AST (U/L)				
ALT (U/L)				
Total bilirubin (mg/dL)				
PLT (10^9^/L)				
INR				

Table shading indicates that the variable has a confounding effect on other factors, and thus was not included in the multivariate analysis. Abbreviations: AFP—alpha-fetoprotein; ALBI—albumin–bilirubin; ALT—alanine aminotransferase; AST—aspartate aminotransferase; BCLC—Barcelona Clinic Liver Cancer staging; CI—confidence interval; EHS—extrahepatic spread; FIB-4—fibrosis index based on four factors; INR—international normalized ratio; MVI—macrovascular invasion; PLT—platelets; PFS—progression-free survival.

**Table 4 cancers-13-03758-t004:** Results of multivariate Cox regression analyses of predictors of OS after regorafenib therapy.

	ALBI Grade-Based Model		Combined ALBI and AFP-Based Model	
Variables	Multivariate Hazard Ratio (95% CI)	*p* Value	Multivariate Hazard Ratio (95% CI)	*p* Value
Age (year)				
Male vs. female	2.643 (0.584–11.95)	0.207	2.488 (0.558–11.09)	0.232
ALBI grade 1 vs. 2	0.543 (0.190–1.556)	0.256		
FIB-4 < 3.25 vs. ≥3.25	0.514 (0.158–1.671)	0.269	0.638 (0.197–2.066)	0.453
BCLC stage B vs. C				
MVI (no vs. yes)				
EHS (no vs. yes)				
AFP (ng/mL) <20 vs. ≥20 (ng/mL)	0.322 (0.109–0.946)	0.039		
ALBI grade 2 and AFP ≥ 20 ng/mL (yes vs. no)			3.783 (1.316–10.88)	0.014
Albumin (g/dL)				
AST (U/L)				
ALT (U/L)				
Total bilirubin (mg/dL)				
PLT (10^9^/L)				
INR				

Table shading indicates that the variable has a confounding effect on other factors, and thus was not included in the multivariate analysis. Abbreviations: AFP—alpha-fetoprotein; ALBI—albumin-bilirubin; ALT—alanine aminotransferase; AST—aspartate aminotransferase; BCLC—Barcelona Clinic Liver Cancer staging; CI—confidence interval; EHS—extrahepatic spread; FIB-4—fibrosis index based on four factors; INR—international normalized ratio; MVI—macrovascular invasion; OS—overall survival; PLT—platelets.

**Table 5 cancers-13-03758-t005:** Results of multivariate Cox regression analyses of predictors of OS after sorafenib-regorafenib sequential therapy.

	ALBI Grade-Based Model		Combined ALBI and AFP-Based Model	
Variables	Multivariate Hazard Ratio (95% CI)	*p* Value	Multivariate Hazard Ratio (95% CI)	*p* Value
Age (year)				
Male vs. female				
ALBI grade 1 vs. 2	0.303 (0.112–0.821)	0.019		
FIB-4 < 3.25 vs. ≥3.25	0.548 (0.161–1.864)	0.335	0.648 (0.181–2.316)	0.504
BCLC stage B vs. C	0.612 (0.133–2.814)	0.528	0.790 (0.139–4.485)	0.790
MVI (no vs. yes)				
EHS (no vs. yes)	0.705 (0.205–2.421)	0.579	0.569 (0.129–2.513)	0.457
AFP (ng/mL) <20 vs. ≥20 (ng/mL)	0.335 (0.122–0.919)	0.034		
ALBI grade 2 and AFP ≥ 20 ng/mL (yes vs. no)			4.603 (1.386–15.29)	0.013
Albumin (g/dL)				
AST (U/L)				
ALT (U/L)				
Total bilirubin (mg/dL)				
PLT (10^9^/L)				
INR				

Table shading indicates that the variable has a confounding effect on other factors, and thus was not included in the multivariate analysis. Abbreviations: AFP—alpha-fetoprotein; ALBI—albumin-bilirubin; ALT—alanine aminotransferase; AST—aspartate aminotransferase; BCLC—Barcelona Clinic Liver Cancer staging; CI—confidence interval; EHS—extrahepatic spread; FIB-4—fibrosis index based on four factors; INR—international normalized ratio; MVI—macrovascular invasion; OS—overall survival; PLT—platelets.

**Table 6 cancers-13-03758-t006:** Sorafenib–regorafenib adverse event profile.

**AEs, *n* (%)**	**Sorafenib** **(*n* = 88)**			**Regorafenib** **(*n* = 88)**		
	**Any**	**Gr 3**	**Gr 4**	**Any**	**Gr 3**	**Gr 4**
All TEAE	56 (63.6)	26 (29.5)	1 (1.1)	78 (88.6)	13 (14.8)	1 (1.1)
Drug-related TEAE						
Hand-foot skin reaction	52 (59.1)	19 (21.6)	0	28 (31.8)	9 (10.2)	0
Diarrhea	32 (36.4)	7 (8.0)	1 (1.1)	34 (38.6)	5 (5.7)	0
Hypertension	13 (14.8)	1 (1.1)	0	4 (4.5)	0	0
Skin rash	11 (12.5)	1 (1.1)	0	5 (5.7)	0	0
Fatigue	8 (9.1)	0	0	5 (5.7)	0	0
Hair loss	5 (5.7)	0	0	0	0	0
Decreased appetite	4 (4.5)	0	0	4 (4.5)	0	0
Abdominal pain	2 (2.3)	0	0	5 (5.7)	0	0
Abnormal LFTs	1 (1.1)	0	0	4 (4.5)	0	1 (1.1)
**AE, *n* (%)**	**Hand-foot Skin Reaction (During Regorafenib Therapy) (*n* = 88)**					
	**Gr 0**	**Any AE**	**Gr 1**	**Gr 2**	**Gr 3**	**Gr 4**
Hand-foot skin reaction (during sorafenib therapy)						
Gr 0, *n* = 36 (100)	29 (80.6)	7 (19.4)	2 (5.6)	1 (2.8)	4 (11.1)	0
Any AE, *n* = 52 (100)	31 (59.6)	21 (40.4)	6 (11.5)	10 (19.2)	5 (9.6)	0
Gr 1, *n* = 10 (100)	9 (90)	1 (10)	1 (10)	0	0	0
Gr 2, *n* = 23 (100)	13 (56.5)	10 (43.5)	3 (13.0)	5 (21.7)	2 (8.7)	0
Gr 3, *n* = 19 (100)	9 (47.4)	10 (52.6)	2 (10.5)	5 (26.3)	3 (15.8)	0
Gr 4, *n* = 0	0	0	0	0	0	0

Abbreviations: AE—adverse event; Gr—group; LFT—liver function test; TEAE—treatment-emergent adverse event.

## Data Availability

The data presented in this study are available from the corresponding author upon reasonable request.

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
