# Peer review of "On-Treatment Albumin-Bilirubin Grade: Predictor of Response and Outcome of Sorafenib-Regorafenib Sequential Therapy in Patients with Unresectable Hepatocellular Carcinoma"

_cancers, 2021, doi:10.3390/cancers13153758_

Round 1
Reviewer 1 Report
The author evaluated the predictors of response and outcome of sorafenib-regorafenib sequential therapy in patients with unresectable hepatocellular carcinoma. We are sorry to inform you that it did not receive a high enough priority for publication in this presentation.
1.There are some reports about the relationship between ALBI grade and prognosis of regorafenib therapy.
Tokunaga T, et al. Int J Clin Oncol 2021;26(5):922-932.
Takada H, et al. Cancers (Basel). 2019;11(9):1256
2.Please state the rationale for determining the cutoff value of AFP and FIB4.
3.Please add values that are not significantly different in multivariate analysis.
Author Response
Response to Reviewer 1 Comments
Thank you very much for your valuable comments on our manuscript (Cancers-1270378). Revisions in the manuscript are shown in red text. Below is the point-by-point response to the specific comments raised by the reviewer and the changes made.
Point 1: There are some reports about the relationship between ALBI grade and prognosis of regorafenib therapy.
- Tokunaga T, et al. Int J Clin Oncol 2021;26(5):922-932.
- Takada H, et al. Cancers (Basel). 2019;11(9):1256 

Response 1:
How to choose the right patient or optimal drug for the second-line sequential therapy remains an unmet issue and needs more evidence or real-word experience in different ways of sequential therapy. Takada et al. demonstrated that highly preserved liver function (Child-Pugh A5 or ALBI grade 1) after failure of sorafenib treatment is beneficial for second- and third-line sequential therapy in patients with unresectable HCC. Tokunaga et al. reported that baseline modified ALBI grade in combination with Child-Pugh score might be a predictor of response to second-line sequential therapy, including regorafenib or ramucirumab for sorafenib-treated HCC patients. We have added these two references to the Introduction to highlight that ALBI grade is used to represent liver reserve and it could be the predictor of treatment outcome or one of the selection criteria for receiving second-line sequential therapy. Focusing on the sorafenib–regorafenib sequential therapy, we aimed to investigate predictors of regorafenib treatment response and survival outcome. We have revised the Introduction to emphasize the purpose of the present study.
Change: We have revised the content under Introduction. (Page 2, Lines 54-56 and Lines 61-71).
Point 2: Please state the rationale for determining the cutoff value of AFP and FIB4.
Response 2:
We used a cutoff value of 3.25 for FIB-4 to differentiate advanced fibrosis or not. A FIB-4 cutoff value of > 3.25 predicted advanced fibrosis with a specificity of 97% and a positive predictive value of 65% [1]. We used AUROC to analyse the predictive performance of AFP values for overall survival and the AUROC was 0.669. Then, we determined the optimal cutoff value according to the Youden index. The optimal cutoff value was 17.3 ng/mL with a sensitivity of 77.8% and a specificity of 59.4%. We also used AUROC to analyse the predictive performance of AFP values for PFS. The AUROC was 0.629. We determined the optimal cutoff value based on the Youden index. The optimal cutoff value was also 17.3 ng/mL with a sensitivity of 56.3% and a specificity of 73.9%. Therefore, we adopted the popular integer value 20 ng/mL as the cutoff value of AFP for further analysis.
Reference
- Sterling RK, Lissen E, Clumeck N, et. al. Development of a simple noninvasive index to predict significant fibrosis patients with HIV/HCV co-infection. Hepatology 2006; 43:1317-1325.
Change: We have revised the content under Materials and Methods. (Page 2, Lines 92-93 and Page 3, Lines 103-107).
Point 3: Please add values that are not significantly different in multivariate analysis.
Response 3:
We have added P values that are not significantly different in multivariate analysis (Revisions are shown in red text in all tables, supplementary tables and manuscript).

Reviewer 2 Report
This study by Wang et al. is of quite interest for scientist working in the field of liver diseases. Here are my comments to improve the quality of this manuscript:
- English should be revised for some sentences due to minor mistakes found
- Introduction should be more detailed - that would allow the readers to better emphasize on the presented work.
- In the Discussion part, the paragraph describing the different limitations of this study should be separated from the paragraph explaining why this study is relevant.
- The fact that patients may have received other therapies in addition to sequential therapy (ST) may constitue an issue in my opinion because of their effect on ST, even if similar proportions of untreated patients also received these types of therapies - can the authors better prove that the ST-derived effects observed on patients are really independent of other therapies ?
- Conclusion is too short - a conclusion part should allow the readers understand how the described work presented in an article add a missing piece to the existing puzzle - here, it is not the situation so please modify it.
Author Response
Response to Reviewer 2 Comments
Thank you very much for your valuable comments on our manuscript (Cancers-1270378). Revisions in the manuscript are shown in red text. Below is the point-by-point response to the specific comments raised by the reviewer and the changes made.
Point 1: English should be revised for some sentences due to minor mistakes found
Response 1:
Our manuscript has been edited by Valery Noah (Wallace Academic Editing, Taiwan) to improve English usage to the best of the editor's ability and we have provided the English Editing Certificate as an attached supplementary file for your reference.
Point 2: Introduction should be more detailed - that would allow the readers to better emphasize on the presented work.
Response 2:
We have revised the Introduction to emphasize the purpose of the present study, which is to address the current unmet issue in our patient cohort with sorafenib–regorafenib sequential therapy.
Change: We have revised the content under Introduction. (Page 2, Lines 54-56; 61-71).
Point 3: In the Discussion part, the paragraph describing the different limitations of this study should be separated from the paragraph explaining why this study is relevant.
Response 3:
We have revised our Discussion and the paragraph describing the different limitations of this study has been separated from the paragraph explaining why this study is relevant.
Change: We have revised the content under Discussion. (Page 12, Lines 330-347).
Point 4: The fact that patients may have received other therapies in addition to sequential therapy (ST) may constitute an issue in my opinion because of their effect on ST, even if similar proportions of untreated patients also received these types of therapies - can the authors better prove that the ST-derived effects observed on patients are really independent of other therapies?
Response 4:
In order to further evaluate the effect of the sequential therapy, we combined these two patient cohorts with or without sequential therapy (n = 178) to investigate the predictors of overall survival. In the ALBI grade-based model, ALBI grade (1 vs. 2), AFP (< 20 vs. ≥ 20 ng/mL), sequential therapy (yes vs. no) and locoregional therapy (yes vs. no) were independent predictors for overall survival after failure of sorafenib therapy (Table S8). In the combined ALBI and AFP-based model, ALBI grade 2 and AFP ≥ 20 ng/mL (yes vs. no), sequential therapy (yes vs. no) and locoregional therapy (yes vs. no) were independent predictors for overall survival after failure of sorafenib therapy (Table S8). To eliminate the confounding effect of locoregional therapy, the subgroup of patients without receiving locoregional therapy after failure of sorafenib therapy (n = 78) was selected to analyze the effect of the sequential therapy. For overall survival after failure of sorafenib therapy, AFP (< 20 vs. ≥ 20 ng/mL), sequential therapy (yes vs. no) were independent predictors in the ALBI grade-based model. ALBI grade 2 and AFP ≥ 20 ng/mL (yes vs. no), sequential therapy (yes vs. no) were independent predictors in the combined ALBI and AFP-based model (Table S9). For overall survival after sorafenib therapy, sequential therapy (yes vs. no) was also an independent predictor. Thus, sequential therapy was an independent predictor of OS not only for the entire cohort but also for the subgroup without receiving locoregional therapy after failure of sorafenib therapy (Tables S10 and S11).
Change: We have described these findings under Results. (Page 11, Lines 249-267) and in Supplementary Tables S8 to S11.
Table S8. Results of univariate and multivariate Cox regression analyses of predictors of OS after failure of sorafenib therapy (all patients, n = 178).
Patients with and without regorafenib therapy after failure of sorafenib therapy (total n = 178) |
||||||
|
|
|
ALBI grade-based Model |
Combined ALBI and AFP-based Model |
||
Variables |
Univariate Hazard Ratio (95% CI) |
p value |
Multivariate Hazard Ratio (95% CI) |
p value |
Multivariate Hazard Ratio (95% CI) |
p value |
Age (year) |
0.984 (0.966–1.002) |
0.075 |
|
|
|
|
Male vs female |
1.155 (0.703–1.899) |
0.570 |
|
|
|
|
ALBI grade 1 vs 2 |
0.446 (0.295–0.675) |
< 0.001 |
0.576 (0.371–0.894) |
0.014 |
|
|
FIB-4 < 3.25 vs ≥ 3.25 |
0.982 (0.664–1.452) |
0.927 |
|
|
|
|
BCLC stage B vs C |
0.429 (0.208–0.885) |
0.022 |
0.834 (0.362–1.922) |
0.670 |
0.704 (0.307–1.617) |
0.409 |
MVI (no vs yes) |
0.849 (0.576–1.251) |
0.408 |
|
|
|
|
EHS (no vs yes) |
0.709 (0.462–1.085) |
0.113 |
1.048 (0.630–1.742) |
0.858 |
1.163 (0.707–1.911) |
0.552 |
AFP (ng/mL) < 20 vs ≥ 20 (ng/mL) |
0.466 (0.308–0.705) |
< 0.001 |
0.509 (0.325–0.795) |
0.003 |
|
|
ALBI grade 2 and AFP ≥ 20 ng/mL (yes vs no) |
2.687 (1.826–3.954) |
< 0.001 |
|
|
2.019 (1.338–3.047) |
0.001 |
Albumin (g/dL) |
0.468 (0.321–0.682) |
< 0.001 |
|
|
|
|
AST (U/L) |
1.008 (1.004–1.011) |
< 0.001 |
|
|
|
|
ALT (U/L) |
1.003 (1.000–1.006) |
0.084 |
|
|
|
|
Total bilirubin (mg/dL) |
2.362 (1.518–3.676) |
< 0.001 |
|
|
|
|
PLT (109/L) |
1.000 (0.996–1.004) |
0.940 |
|
|
|
|
INR |
1.820 (1.061–3.121) |
0.030 |
0.790 (0.383–1.626) |
0.522 |
0.841 (0.427–1.659) |
0.618 |
Sequential therapy (sorafenib and regorafenib vs sorafenib alone) |
0.180 (0.109–0.298) |
< 0.001 |
0.201 (0.117–0.345) |
< 0.001 |
0.196 (0.116–0.332) |
< 0.001 |
Locoregional therapy (yes vs no) |
0.724 (0.495–1.060) |
0.097 |
0.572 (0.376–0.871) |
0.009 |
0.563 (0.376–0.842) |
0.005 |
Abbreviations: AFP—alpha-fetoprotein; ALBI—albumin–bilirubin; ALT—alanine aminotransferase; AST—aspartate aminotransferase; BCLC—Barcelona Clinic Liver Cancer staging; CI—confidence interval; EHS—extrahepatic spread; FIB-4—fibrosis index based on four factors; INR—international normalized ratio; MVI—macrovascular invasion; OS—overall survival; PLT—platelets.
Table S9. Results of univariate and multivariate Cox regression analyses of predictors of OS after failure of sorafenib therapy (subgroup without locoregional therapy, n = 78).
Patients without locoregional therapy after failure of sorafenib therapy (n = 78) |
||||||
|
|
|
ALBI grade-based Model |
Combined ALBI and AFP-based Model |
||
Variables |
Univariate Hazard Ratio (95% CI) |
p value |
Multivariate Hazard Ratio (95% CI) |
p value |
Multivariate Hazard Ratio (95% CI) |
p value |
Age (year) |
0.985 (0.960–1.010) |
0.243 |
|
|
|
|
Male vs female |
2.482 (0.979–6.295) |
0.056 |
1.906 (0.660–5.502) |
0.233 |
1.496 (0.570–3.923) |
0.413 |
ALBI grade 1 vs 2 |
0.394 (0.195–0.795) |
0.009 |
0.551 (0.262–1.161) |
0.117 |
|
|
FIB-4 < 3.25 vs ≥ 3.25 |
1.129 (0.628–2.028) |
0.686 |
|
|
|
|
BCLC stage B vs C |
1.024 (0.317–3.309) |
0.968 |
|
|
|
|
MVI (no vs yes) |
0.775 (0.431–1.392) |
0.393 |
|
|
|
|
EHS (no vs yes) |
1.305 (0.685–2.488) |
0.418 |
|
|
|
|
AFP (ng/mL) < 20 vs ≥ 20 (ng/mL) |
0.559 (0.313–0.998) |
0.049 |
0.532 (0.286–0.990) |
0.046 |
|
|
ALBI grade 2 and AFP ≥ 20 ng/mL (yes vs no) |
2.817 (1.589–4.994) |
< 0.001 |
|
|
2.052 (1.102–3.822) |
0.023 |
Albumin (g/dL) |
0.411 (0.237–0.716) |
0.002 |
|
|
|
|
AST (U/L) |
1.017 (1.010–1.025) |
< 0.001 |
|
|
|
|
ALT (U/L) |
1.012 (1.004–1.020) |
0.003 |
|
|
|
|
Total bilirubin (mg/dL) |
2.493 (1.320–4.708) |
0.005 |
|
|
|
|
PLT (109/L) |
0.998 (0.985–1.010) |
0.714 |
|
|
|
|
INR |
1.644 (0.940–2.875) |
0.082 |
0.846 (0.411–1.741) |
0.650 |
0.867 (0.433–1.735) |
0.686 |
Sequential therapy (sorafenib and regorafenib vs sorafenib alone) |
0.181 (0.091–0.358) |
< 0.001 |
0.232 (0.113–0.479) |
< 0.001 |
0.211 (0.104 –0.428) |
< 0.001 |
Abbreviations: AFP—alpha-fetoprotein; ALBI—albumin–bilirubin; ALT—alanine aminotransferase; AST—aspartate aminotransferase; BCLC—Barcelona Clinic Liver Cancer staging; CI—confidence interval; EHS—extrahepatic spread; FIB-4—fibrosis index based on four factors; INR—international normalized ratio; MVI—macrovascular invasion; OS—overall survival; PLT—platelets.
Table S10. Results of univariate and multivariate Cox regression analyses of predictors of OS after sorafenib therapy (all patients, n = 178).
Patients with and without regorafenib therapy after failure of sorafenib therapy (total n = 178) |
||||||
|
|
|
ALBI grade-based Model |
Combined ALBI and AFP-based Model |
||
Variables |
Univariate Hazard Ratio (95% CI) |
p value |
Multivariate Hazard Ratio (95% CI) |
p value |
Multivariate Hazard Ratio (95% CI) |
p value |
Age (year) |
0.986 (0.969–1.002) |
0.092 |
|
|
|
|
Male vs female |
1.200 (0.731–1.970) |
0.471 |
|
|
|
|
ALBI grade 1 vs 2 |
0.499 (0.332–0.749) |
0.001 |
0.740 (0.479–1.144) |
0.176 |
|
|
FIB-4 < 3.25 vs ≥ 3.25 |
1.118 (0.755–1.655) |
0.577 |
|
|
|
|
BCLC stage B vs C |
0.292 (0.142–0.602) |
0.001 |
0.532 (0.231–1.222) |
0.137 |
0.498 (0.218–1.136) |
0.098 |
MVI (no vs yes) |
0.929 (0.634–1.362) |
0.706 |
|
|
|
|
EHS (no vs yes) |
0.511 (0.332–0.786) |
0.002 |
0.929 (0.565–1.529) |
0.773 |
0.953 (0.582–1.561) |
0.849 |
AFP (ng/mL) < 20 vs ≥ 20 (ng/mL) |
0.466 (0.310–0.701) |
< 0.001 |
0.644 (0.416–0.997) |
0.049 |
|
|
ALBI grade 2 and AFP ≥ 20 ng/mL (yes vs no) |
2.339 (1.589–3.441) |
< 0.001 |
|
|
1.504 (0.998–2.264) |
0.051 |
Albumin (g/dL) |
0.501 (0.343–0.732) |
< 0.001 |
|
|
|
|
AST (U/L) |
1.010 (1.007–1.014) |
< 0.001 |
|
|
|
|
ALT (U/L) |
1.006 (1.002–1.009) |
0.002 |
|
|
|
|
Total bilirubin (mg/dL) |
2.576 (1.627–4.080) |
< 0.001 |
|
|
|
|
PLT (109/L) |
1.001 (0.997–1.005) |
0.719 |
|
|
|
|
INR |
2.151 (1.201–3.854) |
0.010 |
1.025 (0.484–2.172) |
0.949 |
1.087 (0.526–2.247) |
0.821 |
Sequential therapy (sorafenib and regorafenib vs sorafenib alone) |
0.153 (0.093–0.252) |
< 0.001 |
0.201 (0.117–0.344) |
< 0.001 |
0.190 (0.113–0.318) |
< 0.001 |
Locoregional therapy (yes vs no) |
0.740 (0.507–1.081) |
0.119 |
0.738 (0.493–1.105) |
0.140 |
0.739 (0.502–1.088) |
0.125 |
Abbreviations: AFP—alpha-fetoprotein; ALBI—albumin–bilirubin; ALT—alanine aminotransferase; AST—aspartate aminotransferase; BCLC—Barcelona Clinic Liver Cancer staging; CI—confidence interval; EHS—extrahepatic spread; FIB-4—fibrosis index based on four factors; INR—international normalized ratio; MVI—macrovascular invasion; OS—overall survival; PLT—platelets.
Table S11. Results of univariate and multivariate Cox regression analyses of predictors of OS after sorafenib therapy (subgroup without locoregional therapy, n = 78).
Patients without locoregional therapy after failure of sorafenib therapy (n = 78) |
|
|
||||
|
|
|
ALBI grade-based Model |
Combined ALBI and AFP-based Model |
||
Variables |
Univariate Hazard Ratio (95% CI) |
p value |
Multivariate Hazard Ratio (95% CI) |
p value |
Multivariate Hazard Ratio (95% CI) |
p value |
Age (year) |
0.985 (0.962–1.009) |
0.232 |
|
|
|
|
Male vs female |
2.583 (1.018–6.552) |
0.046 |
2.125 (0.700–6.448) |
0.183 |
1.811 (0.681–4.815) |
0.234 |
ALBI grade 1 vs 2 |
0.454 (0.225–0.914) |
0.027 |
0.712 (0.329–1.543) |
0.390 |
|
|
FIB-4 < 3.25 vs ≥ 3.25 |
1.234 (0.689–2.211) |
0.480 |
|
|
|
|
BCLC stage B vs C |
0.485 (0.149–1.577) |
0.229 |
0.382 (0.113–1.292) |
0.122 |
0.367 (0.110–1.221) |
0.102 |
MVI (no vs yes) |
1.180 (0.671–2.076) |
0.566 |
|
|
|
|
EHS (no vs yes) |
0.768 (0.406–1.450) |
0.415 |
|
|
|
|
AFP (ng/mL) < 20 vs ≥ 20 (ng/mL) |
0.601 (0.340–1.061) |
0.079 |
0.604 (0.326–1.117) |
0.108 |
|
|
ALBI grade 2 and AFP ≥ 20 ng/mL (yes vs no) |
2.186 (1.236–3.865) |
0.007 |
|
|
1.610 (0.868–2.988) |
0.131 |
Albumin (g/dL) |
0.531 (0.309–0.912) |
0.022 |
|
|
|
|
AST (U/L) |
1.016 (1.007–1.025) |
< 0.001 |
|
|
|
|
ALT (U/L) |
1.011 (1.002–1.019) |
0.012 |
|
|
|
|
Total bilirubin (mg/dL) |
2.664 (1.367–5.194) |
0.004 |
|
|
|
|
PLT (109/L) |
1.001 (0.988–1.013) |
0.902 |
|
|
|
|
INR |
2.002 (1.080–3.711) |
0.027 |
1.199 (0.567–2.535) |
0.635 |
1.253 (0.610–2.575) |
0.539 |
Sequential therapy (sorafenib and regorafenib vs sorafenib alone) |
0.215 (0.110–0.421) |
< 0.001 |
0.258 (0.126–0.527) |
< 0.001 |
0.245 (0.123–0.488) |
< 0.001 |
Abbreviations: AFP—alpha-fetoprotein; ALBI—albumin–bilirubin; ALT—alanine aminotransferase; AST—aspartate aminotransferase; BCLC—Barcelona Clinic Liver Cancer staging; CI—confidence interval; EHS—extrahepatic spread; FIB-4—fibrosis index based on four factors; INR—international normalized ratio; MVI—macrovascular invasion; OS—overall survival; PLT—platelets.
Point 5: Conclusion is too short - a conclusion part should allow the readers understand how the described work presented in an article add a missing piece to the existing puzzle - here, it is not the situation so please modify it.
Response 5:
We have revised our Conclusion and emphasized the findings of what we analysed for the current unmet issue in our patient cohort with sorafenib–regorafenib sequential therapy.
Change: We have revised the content under Conclusion. (Page 13, Lines 355-357).

Round 2
Reviewer 1 Report
The authors revised the manuscript well according to my suggestions.
Reviewer 2 Report
The corrections raised have been done. I endorse publication of the manuscript.